# Regularized Regression: A New Tool for Investigating and Predicting Tree Growth

**Stuart I. Graham [1,2,*], Ariel Rokem [3], Claire Fortunel [4], Nathan J. B. Kraft [5] and Janneke Hille Ris Lambers [1,2]**

[1] Department of Biology, University of Washington, Seattle, WA 98195, USA;
    janneke.hillerislambers@usys.ethz.ch
[2] Institute of Integrative Biology, ETH Zurich, Universitätstrasse 16, 8092 Zürich, Switzerland
[3] Department of Psychology and eScience Institute, University of Washington, Seattle, WA 98195, USA;
    arokem@uw.edu
[4] AMAP (botAnique et Modélisation de l'Architecture des Plantes et des Végétations),
    Université de Montpellier, CIRAD, CNRS, INRAE, IRD, CEDEX 5, 34398 Montpellier, France;
    claire.fortunel@ird.fr
[5] Department of Ecology and Evolutionary Biology, University of California Los Angeles,
    Los Angeles, CA 90095-1606, USA; nkraft@ucla.edu
[*] Correspondence: sgraham3@uw.edu

**Abstract:** Neighborhood models have allowed us to test many hypotheses regarding the drivers of variation in tree growth, but require considerable computation due to the many empirically supported non-linear relationships they include. Regularized regression represents a far more efficient neighborhood modeling method, but it is unclear whether such an ecologically unrealistic model can provide accurate insights on tree growth. Rapid computation is becoming increasingly important as ecological datasets grow in size, and may be essential when using neighborhood models to predict tree growth beyond sample plots or into the future. We built a novel regularized regression model of tree growth and investigated whether it reached the same conclusions as a commonly used neighborhood model, regarding hypotheses of how tree growth is influenced by the species identity of neighboring trees. We also evaluated the ability of both models to interpolate the growth of trees not included in the model fitting dataset. Our regularized regression model replicated most of the classical model's inferences in a fraction of the time without using high-performance computing resources. We found that both methods could interpolate out-of-sample tree growth, but the method making the most accurate predictions varied among focal species. Regularized regression is particularly efficient for comparing hypotheses because it automates the process of model selection and can handle correlated explanatory variables. This feature means that regularized regression could also be used to select among potential explanatory variables (e.g., climate variables) and thereby streamline the development of a classical neighborhood model. Both regularized regression and classical methods can interpolate out-of-sample tree growth, but future research must determine whether predictions can be extrapolated to trees experiencing novel conditions. Overall, we conclude that regularized regression methods can complement classical methods in the investigation of tree growth drivers and represent a valuable tool for advancing this field toward prediction.

**Keywords:** forest plot; inference; interpolation; model selection; neighborhood model; regularization; test set validation; tree growth

## 1. Introduction

Forest plots where the locations of all trees are mapped contain invaluable information for revealing processes of community assembly and dynamics [1,2]. Neighborhood analyses are one particularly common use of such data, and involve modeling some metric of tree performance (usually growth rate) as a function of the species identities, sizes, and other aspects of neighboring trees [3]. These neighborhood models are typically

used for inferential projects, where several model structures that represent different hypotheses of a process of interest (e.g., competition) are fitted to the same dataset, and model selection is used to determine the hypothesis with most support [4]. This approach has allowed many interesting questions to be addressed, including: How are competitive interactions influenced by environmental variables [5]? How might tree performance respond to climate change [6]? How are competitive interactions moderated by niche similarities and hierarchies [7,8]?

Current neighborhood models are structured around a set of empirically supported patterns of tree performance, but this ecological realism has some drawbacks. For example, tree growth is typically modeled as a non-linear function of tree size and the degree of crowding by neighboring trees [4]. These non-linear relationships increase the number of parameters and the opportunity for local optima, resulting in a slow and computationally intensive model fitting process (for a linearized version, see [9,10]). In addition, the need to fit a separate model for each hypothesis to be tested further increases computation time and thereby limits the number of hypotheses compared; a potential problem when considering the many ways in which processes such as competition can be modeled [2]. Another consequence of ecological realism is that the models representing different hypotheses tend not to be fully nested, meaning that information theoretic approaches such as Akaike's information criterion (AIC) must be used for model selection. This is not ideal because AIC penalizes model complexity and can thereby lead to an overly simple final model and bias the conclusions drawn [11].

Regularized regression is an alternative method for neighborhood analysis that avoids many of these drawbacks. The aim of regularization is the same as AIC model selection; to trade off the antithetical aims of penalizing model complexity and accurately fitting the training data. However, in regularized regression, the strength of this trade-off is determined by a regularization parameter, which is estimated through cross-validation. We focus specifically on Least Absolute Shrinkage and Selection Operator (LASSO) regularized regression [12], which penalizes model complexity by shrinking coefficients of unhelpful covariates to zero and thereby conducts variable selection autonomously. The main benefit of regularized regression is speed; as a linear model, it can be fitted very quickly, and autonomous variable selection removes the requirement of fitting multiple models representing different hypotheses. Moreover, regularized regression is highly robust to correlated variables; therefore, it allows a single model to include many manifestations of the same process and drop all but the most influential driver [12]. Importantly, due to its penalization of model complexity, regularization can result in an overly simple final model [11], but it is unclear whether it leads to more bias than AIC. The main drawback of regularized regression is that it requires a linear modeling framework, which may not accurately approximate the inherently non-linear effects on tree growth [6,13]. Overall, regularized regression offers a distinct set of strengths and weaknesses relative to classical neighborhood models.

Another potential use of neighborhood analyses is to predict tree growth beyond the spatial or temporal limits of training data. To the best of our knowledge, neighborhood models have not been used for out-of-sample prediction, but this is likely to change as growing emphasis on dataset publication [14,15] and new software tools [16] increases the availability of mapped forest plot data. Moreover, accurate predictions of tree growth are highly desirable because they could provide insight to current spatial patterns in carbon storage and the effects of land use and climate change on timber production and global carbon cycling. It is important to recognize two levels of prediction: (1) *interpolation* of observations experiencing conditions contained in the training data (e.g., of trees experiencing environmental conditions similar to those of trees used for model fitting); and (2) *extrapolation* to observations experiencing conditions not present in the training data (e.g., novel climates). However, neither the interpolation nor extrapolation ability of neighborhood models has been tested.

Both classical neighborhood models and regularized regression may be well-suited to prediction, but it is unclear which method will be the most accurate. The biggest pitfall in predictive modeling is applying an overly complex model that fits the sample data well, but does not generalize to other data because it includes relationships that are specific to the sample. Both AIC and regularization are designed to avoid this overfitting problem by penalizing model complexity, and therefore classical and regularized regression neighborhood models may produce accurate out-of-sample predictions. The cross-validation model selection approach used to select the regularization parameter also guards against overfitting and is growing in popularity in ecology [17,18]. However, cross-validation has been shown to be asymptotically equivalent to AIC [19] and also requires models to be fitted to many subsets of the data. To ensure that resources are not wasted on applying cross-validation to classical neighborhood models, it is timely to empirically demonstrate the equivalency between AIC and cross-validation model selection.

In this study, we built a novel regularized regression model of tree growth and used it to test hypotheses regarding how tree growth is influenced by the presence and species identity of neighboring trees. This is a particularly interesting test case for a new modeling method because variation in the abundances of different neighbor species leads to an inherently unbalanced sample, which is likely to create challenges in model fitting. To evaluate the accuracy of our model's inferences, we compared them with those of a commonly used neighborhood model ([4] hereafter, likelihood model). In addition, we investigated the ability of both neighborhood models (i.e., regularized regression and likelihood) to interpolate the growth of trees not used in model fitting. We also evaluated the predictive performance of a cross-validated likelihood model to demonstrate the equivalency between AIC and cross-validation model selection [19]. Overall, we found that regularized regression makes similar inferences to the likelihood model, but that the neighborhood model with the most accurate out-of-sample predictions varies between focal tree species.

## 2. Materials and Methods

### 2.1. Tree Growth Data

The data used in this study came from the mature and old-growth conifer forests of Mount Rainier National Park, WA, USA. Mount Rainier is a 4392 m high volcano and covers a large climatic gradient. Increasing elevation is associated with decreasing temperatures and increasing precipitation, although precipitation is considerably reduced on the eastern side of the volcano due to a rain shadow effect. The region experiences a temperate maritime climate with warm, dry summers and cold, wet winters.

We used data collected in 15 forest plots established in 1977 and 1978 as part of the Pacific Northwest Permanent Sample Plot Program [20]. These plots were intentionally located to capture the diversity of climatic conditions on Mount Rainier, and therefore range in elevation from 581 to 1492 m. All plots are 1 ha (100 × 100 m) in size and, at the time of their establishment, all trees with a diameter at breast height (1.37 m above ground level; hereafter DBH) ≥ 15 cm were tagged, identified to species, mapped on a coordinate grid, and had their DBH recorded. Between 25% and 100% of the area of each plot was also designated as a detailed plot where data were collected on all trees with a DBH ≥ 5 cm. Approximately every five years, all plots are revisited to tag new trees meeting the minimum size threshold, document tree mortality, and to re-measure the size of tagged trees, with the most recent census occurring in 2017.

We calculated average annual growth for each tree as the difference in DBH between its earliest and most recent measurement divided by the number of years elapsed between those measurements. The slow growth rates of trees in this harsh high-elevation environment meant that measurement inaccuracies sometimes resulted in biologically impossible negative growth rates; all such trees were excluded from our analysis (1.6% of focal trees). The smaller trees recorded only in the detailed plots (5–15 cm DBH) were included in our analyses as focal trees but excluded as neighbors that could influence the growth of other

focal trees to prevent systematic bias in neighbor interactions between detailed and non-detailed areas of the plots. Of the 17 tree species included in the dataset, we modeled growth for only the six species represented by at least 100 individuals: *Abies amabilis*, *Callitropsis nootkatensis*, *Pseudotsuga menziesii*, *Thuja plicata*, *Tsuga heterophylla*, and *Tsuga mertensiana* (hereafter: ABAM, CANO, PSME, THPL, TSHE, TSME, respectively; see Table S1 for full species list and Table S2 for a summary of how focal species were distributed across sample plots).

For our neighborhood models, we considered all trees growing within 15 m of a focal tree to be that focal tree's neighbors. This neighborhood size is comparable to those used in other studies [4,7] and was found through our own exploratory analyses to result in the best training data fits. To avoid edge effects, all focal trees within 15 m of a forest plot boundary were excluded from our analysis. Each of the remaining focal trees was assigned to one of four test datasets at random, such that 25% of the focal trees in each plot were placed in each test set. We then defined a corresponding training dataset for each test set such that the first training set consisted of all focal trees not in the first test set (75% of all focal trees). All models were fitted to each of the four training sets to assess the robustness of conclusions; test sets were used to evaluate the predictive skill. As in previous studies, we also created separate models for each of our focal species because model parameter values are expected to differ greatly between species.

The data and code underlying all presented analyses are available on Zenodo at doi:10.5281/zenodo.5512791, reference number [21].

### 2.2. Likelihood Model

The likelihood model [4,22] has the following generalized formula:

$$g = g_{max} \times \delta_t \times \gamma_p \times \omega_t, \tag{1}$$

where $g$ is the predicted growth, $g_{max}$ is an estimated maximum potential growth rate in the absence of neighbors, $\delta$ is a size effect, $\gamma$ is a climate effect, and $\omega$ is a crowding effect, with subscripts indicating whether effects vary between focal trees ($t$) or plots ($p$). The size, climate and crowding effects can take any value between 0 and 1; therefore, growth predictions can take any value between 0 and $g_{max}$.

The size effect ($\delta$) accounts for the expectation that trees have an optimal size at which maximum growth occurs and is modeled as a lognormal distribution:

$$\delta_t = \exp\left(-\frac{1}{2}\left(\frac{\log(DBH_t/X_0)}{X_b}\right)^2\right), \tag{2}$$

Parameter $X_0$ specifies the DBH at which maximum growth occurs, and parameter $X_b$ determines the width of the lognormal distribution. This formulation is highly flexible, allowing the relationship between focal growth and size to be monotonically increasing, monotonically decreasing, or non-monotonic.

The climate effect ($\gamma$) accounts for the expectation that growth rates will differ among the plots due to their different climatic conditions. Although there are many climatic variables that differ dramatically along the elevational gradient on which our plots are situated, these variables are strongly correlated [23], and potential evapotranspiration (PET) is informative of growth rates of our focal species in these plots [24]. Consequently, we used PET as the sole abiotic variable in our models and calculated the average annual PET for each plot from the time of plot establishment up until the most recent tree measurements, following the protocol outlined in [24]. The climate effect ($\gamma$) is modeled as a Gaussian distribution:

$$\gamma_p = \exp\left(-\frac{1}{2}\left(\frac{\text{PET}_p - pet_a}{pet_b}\right)^2\right), \tag{3}$$

where $\text{PET}_p$ represents the average annual PET of plot $p$ (where the focal tree resides), $pet_a$ specifies the PET at which maximum growth occurs, and $pet_b$ determines the width

of the Gaussian distribution. As with the size effect, this flexible structure allows the relationship between focal growth and PET to monotonically increase, monotonically decrease, or be non-monotonic.

To incorporate the effects of neighbors on focal tree growth, a neighborhood crowding index (NCI) was calculated for tree $t$ as:

$$NCI_t = \sum_{i=1}^{S} \sum_{j=1}^{n_i} \frac{DBH_{ij}{}^{\alpha}}{Distance_{ij}{}^{\beta}} \times \lambda_i, \tag{4}$$

where $S$ is the number of neighbor species and $n_i$ is the number of trees of species $i$ in focal tree $t$'s neighborhood. This formula reflects the expectation that a neighbor's influence on focal tree growth increases with its size but decreases with its distance from the focal, and the estimated parameters $\alpha$ and $\beta$ allow these relationships to be non-linear. The effect of neighbor size and distance is also multiplied by an estimated interaction coefficient ($\lambda_i$), which takes a value between 0 and 1 and represents the effect of neighbors of species $i$ on the growth of focal trees of the species being modeled.

The crowding effect ($\omega$) is calculated as a negative exponential function of *NCI*:

$$\omega_t = \exp(-C \times NCI_t), \tag{5}$$

where $C$ is an estimated parameter that modulates the growth response of trees to varying *NCI* values.

To make inferences regarding the effects of neighbors on focal growth, we fitted four variations of the likelihood model for each focal species: (1) no interactions—crowding effect ($\omega$) excluded; (2) equivalent interactions—no $\lambda$ parameters included (quantitatively equivalent to a single $\lambda$ with value 1); (3) conspecific vs. heterospecific interactions—two $\lambda$ parameters, one for conspecific neighbors ($\lambda_{con}$) and another for heterospecific neighbors ($\lambda_{het}$); and (4) species-specific interactions—estimated $\lambda_i$ for each neighbor species $i$. In the species-specific interaction models, rare neighbor species were grouped under a single $\lambda_{other}$ parameter. Rare neighbor species (<5% of neighbors in each focal species × training set combination) were defined as those that appeared as neighbors of the focal species fewer than 100 times, when averaged across the four training sets. We elected to use an average instead of specifying rare neighbor species separately for each training set to ensure that the fitted $\lambda_{other}$ parameters could be compared across training sets. The four model structures for each focal species × training set combination were compared using Akaike's information criterion corrected for a low sample size (AICc).

Parameter values were estimated using the simulated annealing algorithm implemented through the *optim* function in the base library of R 4.0.2 [25]. The optimizations were facilitated through the use of advanced computational, storage, and networking infrastructure provided by the Hyak supercomputer system at the University of Washington.

### 2.3. Regularized Regression Model

In our regularized regression model, focal tree growth was modeled as a linear function of: the species identity, size and proximity of neighbors; PET; and the densities of each neighbor species, and all species combined, in the neighborhood. Rare neighbor species (characterized in the same way as for the likelihood model) were assigned a species identity of "other", and the density of "other" was also included in the model. To estimate the effects of neighbor identity, size and proximity in a linear modeling framework, each focal tree–neighbor interaction was treated as an independent observation. This resulted in a design matrix where each focal tree occupied $n$ rows, with $n$ being the number of neighbors in its neighborhood. The growth rate, PET and densities were necessarily identical across all $n$ rows corresponding to the same focal tree. This design matrix structure resulted in $n$ growth predictions of each focal tree based on each of its $n$ interactions and we used the arithmetic average of these predictions as the final prediction of focal growth.

To meet the assumptions of linear regression, the growth rate data were transformed to approximate a normal distribution as follows:

$$g = \sqrt{\frac{average\ annual\ diameter\ growth}{initial\ DBH}}, \tag{6}$$

This transformation has the additional benefit of partially accounting for the nonlinear relationship between focal size and focal growth included in the likelihood model; it allows a saturating but always monotonic relationship between growth rate and tree size.

We fitted the regularized regression models using the *cv.glmnet* function of the glmnet R package [26]. This function estimates parameter values through a stochastic cyclical coordinate descent algorithm using a set of values for the regularization parameter. It then uses 10-fold cross-validation to evaluate the models fitted with different regularization parameter values, reporting the mean square error (MSE) for each. Of the multiple output models, glmnet indicates the one with the highest regularization value (strongest regularization) that resulted in an MSE within one standard error of the model with the lowest MSE; we used this model for interpretation. In addition, the rapid fitting of the regularized regression models allowed us to fit 100 models for each focal species by training set combination to evaluate how consistent the findings of this model are in the face of the stochastic fitting process. Of these 100 models, the one with the lowest MSE was used for evaluating model fit to the training data and predictive performance. The model fitting procedure implemented by glmnet is rapid; therefore, it was conducted on a personal laptop.

### 2.4. Comparing Inferential Performance

To determine whether our regularized regression model could replicate the inferences of the likelihood model, we compared the conclusions each of the models would have led us to for four commonly asked questions regarding the impact of neighbors on tree growth. Separately for each focal species, we asked: (1) Is focal growth influenced by neighboring trees?; (2) Is focal growth influenced by neighbor species identity?; (3) Is focal growth higher in the presence of conspecific or heterospecific neighbors?; and (4) Which neighbor species are associated with the highest/lowest focal growth? For each modeling approach, the conditions under which we drew particular conclusions regarding these questions are outlined in Table 1.

**Table 1.** Conditions used to draw conclusions regarding inferential questions in the likelihood and regularized regression models.

| Conclusion | Condition | |
|---|---|---|
| | **Likelihood** | **Regularized Regression [1]** |
| Focal growth is influenced by neighboring trees | Best model is: equivalent, conspecific vs. heterospecific, or species-specific | At least one species identity, size, proximity, or density variable retained |
| Focal growth is influenced by neighbor species identity | Best model is: conspecific vs. heterospecific or species-specific | At least one species identity or species-specific density variable retained |
| Focal growth is higher in the presence of conspecifics | In the best conspecific vs. heterospecific model, $\lambda_{het} - \lambda_{con} > 0$ | Coefficient of: neighbor species = focal species and/or focal species density $> 0$ [2] |
| Neighbor species X is associated with: (1) high; (2) medium; (3) low focal growth | In the best species-specific model: (1) $\lambda_X > 0.66$, (2) $0.33 \le \lambda_X \le 0.66$, (3) $\lambda_X < 0.33$ | Coefficient of neighbor species = X is: (1) positive, (2) dropped from model, (3) negative [3] |

relative to the average neigh-
bor

[1] Conclusions were drawn separately for each of the 100 regularized regression models run for each focal species × training set combination. [2] If coefficients of neighbor species = focal species and focal species density had opposite signs, we concluded that focal growth was unaffected by whether neighbors were conspecific or heterospecific in that particular model run. [3] To enable comparison with $\lambda_x$ in the likelihood model, we calculated the number of the 100 regularized regression models where the coefficient of neighbor species = X was positive minus the number where this coefficient was negative to obtain a number between −100 and +100, then rescaled these values to a range of 0–1.

### 2.5. Evaluating Predictive Performance

To investigate the predictive potential of neighborhood models, we evaluated the out-of-sample predictive ability for three different models for each focal species by training set combination: regularized regression, AIC likelihood and CV likelihood. Each of these predictive models was one of the models described in Methods: Inference. For the regularized regression model, we used the model with the lowest MSE (out of the 100 models run). For the AIC and CV likelihood models, we used the models with the lowest AIC and lowest cross-validated MSE, respectively, of the four model structures. To identify the lowest cross-validated MSE, we divided each training set into 10 folds (consistent with regularized regression cross-validation), fitted each of the four likelihood model structures to each possible set of 9 folds, then calculated MSE of the models' predictions of the 10th fold. We averaged the resulting 10 MSE values to obtain the cross-validated MSE of each model structure.

We quantified the out-of-sample prediction (interpolation) ability of each model type applied to each training set by calculating the coefficient of determination ($R^2$) of the model when applied to its corresponding test set, which was entirely unused in the fitting of that model (see Section 2.1 Tree growth data for how training and test sets were defined). This metric measures the proportion of variance around the mean value of the dependent variable explained by the model. The maximum possible value for a coefficient of determination is 1 (all variance explained), but negative values can exist when a model is applied to unseen test data if there is more unexplained variation around model predictions than exists around the mean growth value in the test data.

It is often advised that training and test sets be spatially or temporally separated to prevent overestimates of predictive ability that can result from spatial or temporal autocorrelation. In this study, we present results from spatially overlapping training and test sets because the predictive performance of the models was similar when the spatially separated training and test sets were used. Moreover, although we suspect that spatial and temporal autocorrelation in unmeasured variables which influence tree growth was likely in our dataset (e.g., soil conditions, pest damage), we do not know the scale of such variation—which means that it is unclear whether spatial or temporal separation would address such autocorrelation.

## 3. Results

### 3.1. Comparing Inferential Performance

The regularized regression models generally led to the same qualitative conclusions regarding tree growth hypotheses, as did the AIC likelihood models, but did so in a fraction of the time. In combination, the regularized regression models took <15 min to fit on a personal laptop, whereas the AIC likelihood models took 339 h on the HYAK supercomputer system. The AIC likelihood models were quite consistent across training sets with regard to the best-fitting model structure (Table S3) and fitted parameters (Tables S4–S9; Figures S1 and S2). The regularized regression models also showed consistency in the variables retained and parameter values (Tables S10–S15). The two modeling methods agreed that growth in all focal species was influenced by both neighboring trees and their

species identity in all training sets, with the one exception of the likelihood model finding PSME growth to be unaffected by neighbor species identity in one training set. There was also overall agreement between methods on whether focal growth was higher in the presence of conspecific or heterospecific neighbors (Table 2). With perfect consistency across training sets, both methods found ABAM to grow faster in the presence of conspecific neighbors, but found CANO, TSHE, and TSME to grow faster in the presence of heterospecific neighbors. The regularized regression found conspecific and heterospecific neighbors to be associated with equal focal tree growth rates in all PSME training sets and three of the THPL training sets. Somewhat consistent with this, the conclusions of the AIC likelihood model for PSME and THPL changed the direction across training sets (Table S16).

**Table 2.** Is focal growth higher in the presence of conspecific or heterospecific neighbors? Higher growth in the presence of conspecifics represents positive feedback on growth and is indicated with '+'. Negative feedback and the absence of feedback are indicated with '−' and '0', respectively. NA values for regularized regression indicate that the model did not find growth to be substantially higher in the presence of conspecifics or heterospecifics. See Table S16 for numerical outputs.

| Focal Species | Likelihood | | | | Regularized Regression | | | |
|---|---|---|---|---|---|---|---|---|
| | 1 | 2 | 3 | 4 | 1 | 2 | 3 | 4 |
| ABAM | + | + | + | + | + | + | + | + |
| CANO | − | − | − | − | − | − | − | − |
| PSME | + | − | − | − | NA | NA | NA | NA |
| THPL | − | − | 0 | − | NA | − | NA | NA |
| TSHE | − | − | − | − | − | − | − | − |
| TSME | − | − | − | − | − | − | − | − |

The regularized regression and AIC likelihood methods showed less agreement on which neighbor species were associated with the highest and lowest focal growth (Figures 1 and S3–S7). Some examples of agreement were that ABAM grew quickly in the presence of conspecific neighbors, but slowly in the presence of THPL and TSME neighbors (Figure 1), and TSHE grew slowly in the presence of CANO and PSME neighbors (Figure S6). The most common scenario of disagreement was when the AIC likelihood model found a neighbor species to be associated with high or low focal growth, but the regularized regression defaulted to medium (neighbor identity variable dropped from model). The likelihood model sometimes concluded that a neighbor species was associated with low growth of a particular focal species in one training set but high growth of that same focal species in another training set (e.g., the effect of PSME on focal ABAM; Figure 1); in contrast, regularized regression was much more consistent across training sets. The only case of direct disagreement was in the effect of ABAM on focal PSME (Figure S4).

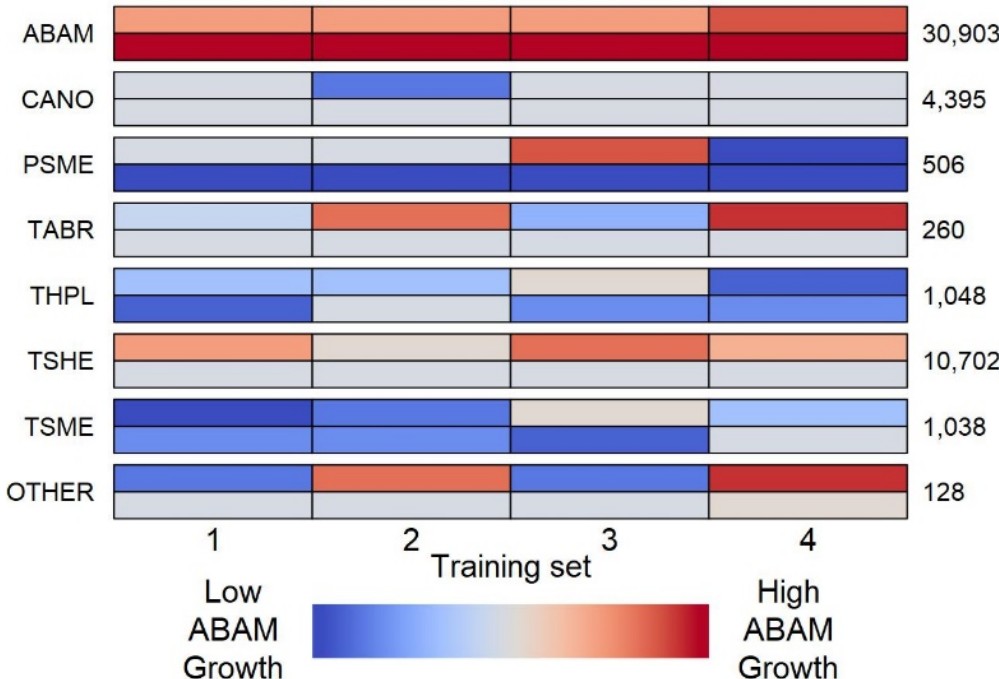

**Figure 1.** Which neighbor species are associated with the highest/lowest growth of ABAM focals? For each neighbor species, there are two rows of colored bars. The top row of bars shows the likelihood model results, and the bottom row shows the regularized regression model results. Each row of bars is divided into four sections, to show the results according to the models fitted to each of the four training sets. The color of the bars indicates the growth rate of ABAM in the presence of the neighbor species that row represents (see inset legend). The numbers on the right of the figure indicate the number of neighbors of each species averaged across training sets.

### 3.2. Evaluating Predictive Performance

The modeling method that resulted in the highest out-of-sample predictive skill differed between focal species (Figure 2b). The likelihood models (AIC and CV) made more accurate predictions than the regularized regression models for ABAM and THPL, but regularized regression performed best for PSME. All three methods had similar predictive skill for CANO, TSHE and TSME, although only regularized regression maintained some predictive skill (coefficient of determination > 0) for TSME in all training sets. Importantly, the fit to training data (Figure 2a) was often considerably higher than the fit to out-of-sample data (Figure 2b; e.g., CANO and TSME). The AIC and CV likelihood models had almost identical out-of-sample predictive skill (Table S17); however, the CV model required far more computation (881 h on the high-performance computing cluster). The final CV likelihood model often had a different structure from the final AIC likelihood model, but was not consistently more or less complex (Tables S3 and S18).

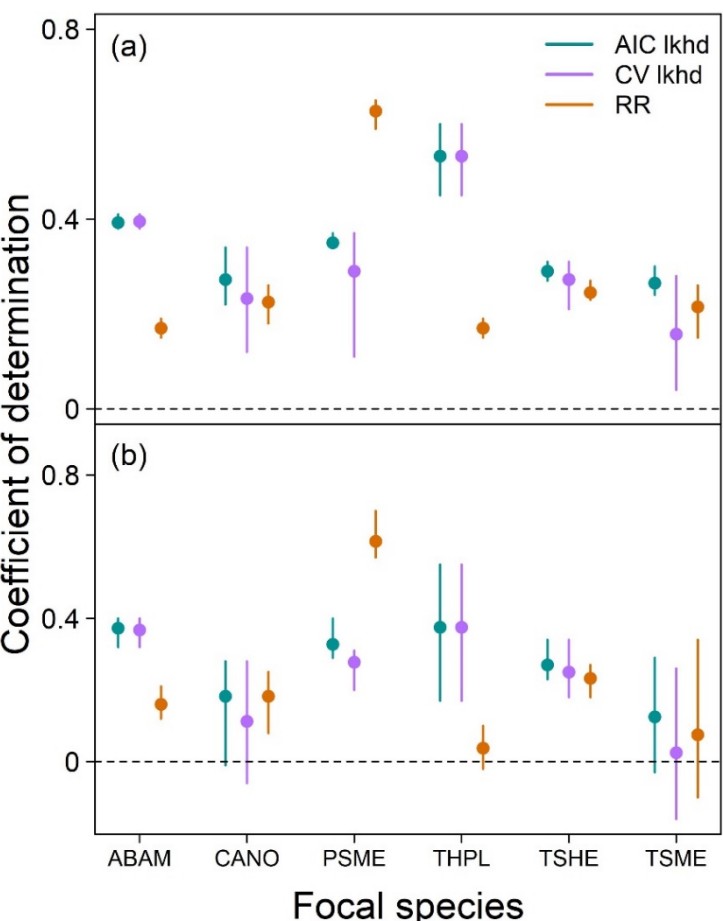

**Figure 2.** Training fit (**a**) and out-of-sample predictive skill (**b**) of the likelihood and regularized regression models. Likelihood models using AIC and cross-validation for model selection are shown in green and purple, respectively. Regularized regression models are shown in orange. Points and error bars represent the mean and range of coefficients of determination across the four training sets. Raw values are provided in Table S17.

## 4. Discussion

We found that our inferential conclusions regarding the effects of neighbors would have been very similar whether using the regularized regression model or the more realistic likelihood model of tree growth. In contrast, the method that led to the most accurate out-of-sample predictions varied between focal species. The biggest difference between the two methods was in the time taken for model fitting, with the regularized regression taking minutes on a personal computer and the likelihood approach requiring many hours on a computing cluster. In all, we believe the two approaches to each have their advantages, and we encourage further investigation of how novel regularized regression can complement the classical likelihood method.

### 4.1. Using Regularized Regression for Inference

The regularized regression model generally reached the same conclusions as the likelihood model regarding hypotheses for how focal tree growth is affected by neighbors. The two modeling approaches were in almost total agreement that the growth of all focal species was influenced by neighbors and their species identities. We have confidence in these conclusions because they align with previous studies which found tree growth to be influenced by: (1) the local neighborhood in these plots [24,27]; and (2) neighbor species identity in similar forests of British Columbia [28,29]. The two modeling approaches also agreed that focal trees of ABAM grew more quickly in the presence of conspecifics, but

that the opposite was true for CANO, TSHE, and TSME (Table 2). Furthermore, regularized regression concluded that the growth of PSME and THPL was similar in the presence of conspecific and heterospecific neighbors, which is consistent with the likelihood model's findings of small and variable signals across training sets for PSME and THPL, respectively (Table S16). The positive and negative feedbacks on the growth of ABAM and TSHE, respectively, and the similar effects of conspecific and heterospecific neighbors on THPL, do have some support in the literature [28,29], but we are not aware of previous assessments for CANO, TSME, or PSME. The agreement between regularized regression and likelihood models was lower when considering which neighbor species were associated with high/low focal growth (Figures 1 and S3–S7). Together, our results demonstrate strong agreement in the inferences drawn from regularized regression and likelihood models, particularly in broad-scale patterns, and it is unclear whether the generally more complex likelihood models represent a lack of sensitivity of regularized regression or false detection by the likelihood model.

Our novel regularized regression approach may be particularly well-suited to the inferential analysis of large data sets due to its low computational requirements and automation of model selection. By transforming the growth variable to account for its non-linear relationship with focal tree size, we constructed an accurate linear model that could be fitted in minutes on a personal computer. However, it is certainly possible that in datasets where the degree of non-linearity is more extreme, linearizing transformations will be insufficient [13]. For example, our growth transformation does not permit a decrease in growth rates at larger tree sizes, an often-documented pattern that was very weak in our dataset. It should be noted that linearization has been applied to classical neighborhood models, thereby reducing their computational requirements [9,10], but these models still fit far more slowly than regularized regression and do not automate model selection. In a likelihood modeling framework, a separate model representing each of the several hypotheses to be tested must be fitted independently, greatly increasing the total model-fitting time and constraining possible conclusions to those permitted by comparisons among the finite set of constructed models. This can be problematic due to the many different ways there are to model processes such as competition [2]. In contrast, an almost-unlimited number of variables pertaining to different models of competition (e.g., measures of functional and phylogenetic distances) can be included in a single regularized regression model, and those that have the greatest influence will be indicated. This model selection capability suggests that regularized regression could also be used as an exploratory tool prior to developing a likelihood model by, for example, selecting among the many potential climate variables to include.

### 4.2. Using Neighborhood Models for Prediction

The absolute and relative abilities of the likelihood and regularized regression methods to predict out-of-sample data are likely to vary among species and systems. Both methods demonstrated some out-of-sample predictive ability for all focal species in at least some training sets, although this ability was weak and variable for TSME, the focal species with the lowest sample size (Figure 2). The likelihood model, whether using AIC or cross-validation for model selection, more accurately predicted the growth of ABAM and THPL, whereas the regularized regression made better predictions for PSME. The high predictive skill of the regularized regression model for PSME may be due to the apparently small effect of neighbors on this species' growth, indicated by the small fitted value of parameter C in the likelihood model (Table S6). This suggests that PSME growth can be predicted using only the focal tree size and PET, which may be reasonably approximated with linear relationships (Figures S1 and S2). Similarly, the higher predictive ability of the likelihood model for ABAM could be a result of the highly non-linear PET effect observed in this species (Figure S2). We also found the regularized regression method to show more consistent predictive ability across training sets, as evidenced by it being the only method to maintain predictive skill across all training sets for CANO (Figure 2b). As

was predicted from the proven equivalency of AIC and cross-validation [19], the predictive performance of the likelihood model did not increase when the more computationally intensive cross-validation technique was used for model selection.

Further research is certainly needed before neighborhood models can be used for prediction purposes, but our results do provide some important lessons. We have shown that both the likelihood and regularized regression models can usually interpolate the growth rates of trees experiencing similar conditions to those in the training set. However, a test of whether these models' predictions can be extrapolated to trees experiencing different conditions, such as novel climates, is still lacking, and we encourage research in this area. We also learned that the inclusion of neighbor interactions generally improves the accuracy of tree growth predictions, as evidenced by the results of our AIC and cross-validation model selection that are designed for predictive modeling. Additionally, our finding of poor out-of-sample predictive ability for some focal species (e.g., TSME) reinforces the practice of always evaluating predictive models using a test set [11]. Although setting aside 20% to 25% of a hard-earned dataset for testing can be unsettling, we argue that this practice is critical, especially where results may influence stakeholder decision-making. Moreover, ecological datasets are quickly growing larger and more accessible, thereby enabling us to alter best practices as our ability to address many important questions in the ecological field becomes less limited by data. Another lesson learned is that the most appropriate neighborhood modeling method will vary among species, with potentially large effects on predictive performance (e.g., PSME; Figure 2b). We advise consideration of whether linearizing transformations are appropriate for a given dataset before deciding whether to use regularized regression vs. likelihood approaches.

Regularized regression models may be particularly useful for predictive projects due to their rapid fitting and ability to include many correlated explanatory variables. Most predictive models are intended for application over large spatial areas, and consequently will be trained on large datasets. As a result, models that can be fitted quickly may be necessary, and regularized regression has a clear advantage over likelihood approaches in this regard, even when likelihood models are linearized [9]. We also argue that the full potential of regularized regression models is far from realized. Our regularized regression models accounted only for the non-linear relationship between growth and focal size, but it may be possible to use other transformations to accommodate other complex relationships. For example, the influence of a neighboring tree is expected to vary non-linearly with that neighbor's DBH—this could be incorporated in a linear framework by using: (1) $\sqrt{DBH}$; (2) $DBH^2$; (3) only neighbors with DBH > focal DBH, etc. In a likelihood framework, a separate model would need to be fitted for each of these neighbor DBH transformations, whereas a regularized regression could incorporate all of them at once and indicate the most informative, which could then optionally be used to design a likelihood model.

## 5. Conclusions

We have developed a regularized regression model of neighborhood-dependent tree growth that can replicate the ecological inferences of a classical likelihood model in a fraction of the time. Regularization is particularly efficient for inferential projects because it automates the process of model selection and can handle correlated explanatory variables. This feature means that our regularized regression model could also be used to select among potential explanatory variables (e.g., climate variables) and thereby streamline the development of a classical likelihood model. We encourage the investigation of regularization as a tool for modeling tree growth and other processes that have many potential covariates, such as seedling survival [30] and mature tree mortality [31].

We have also shown that neighborhood models, including regularized regression, can provide accurate growth predictions of trees not used in model fitting. However, we only tested the model's predictive skill on trees that experienced similar conditions to those used in model fitting. Future research should investigate whether the findings of

neighborhood models can be extrapolated to trees experiencing conditions absent from the training set (e.g., novel climates). Although the optimal neighborhood modeling approach for prediction will vary among species and systems, we believe that our regularized regression has great potential due to its rapid fitting and ability to include many explanatory variables that represent different models of complex processes such as competition. Overall, we found that regularized regression and likelihood approaches are complementary to better understand the drivers of tree growth, and suggest that regularization will be a valuable tool for advancing the field of tree growth modeling toward prediction.

**Supplementary Materials:** The following are available online at www.mdpi.com/article/10.3390/f12091283/s1, Supporting Tables: Table S1. Species included in neighborhood analyses as either focal trees or neighbors. Table S2. Abundance, size, and density of focal species across sampled plots. Table S3. AIC likelihood model selection results. Table S4. Fitted parameter values for AIC likelihood models of ABAM. Table S5. Fitted parameter values for AIC likelihood models of CANO. Table S6. Fitted parameter values for AIC likelihood models of PSME. Table S7. Fitted parameter values for AIC likelihood models of THPL. Table S8. Fitted parameter values for AIC likelihood models of TSHE. Table S9. Fitted parameter values for AIC likelihood models of TSME. Table S10. Estimated coefficients for regularized regression models of ABAM. Table S11. Estimated coefficients for regularized regression models of CANO. Table S12. Estimated coefficients for regularized regression models of PSME. Table S13. Estimated coefficients for regularized regression models of THPL. Table S14. Estimated coefficients for regularized regression models of TSHE. Table S15. Estimated coefficients for regularized regression models of TSME. Table S16. Is focal growth higher in the presence of conspecific or heterospecific neighbors? Table S17. Amount of variance in the training and test datasets explained by the AIC likelihood (L-AIC), CV likelihood (L-CV) and regularized regression (RR) models. Table S18. CV likelihood model selection results. Supporting Figures: Figure S1. Fitted relationships between focal tree size and growth for the AIC likelihood models. Figure S2. Fitted relationships between potential evapotranspiration (PET) and growth for the AIC likelihood models. Figure S3. Which neighbor species are associated with the highest/lowest growth of CANO focals? Figure S4. Which neighbor species are associated with the highest/lowest growth of PSME focals? Figure S5. Which neighbor species are associated with the highest/lowest growth of THPL focals? Figure S6. Which neighbor species are associated with the highest/lowest growth of TSHE focals? Figure S7. Which neighbor species are associated with the highest/lowest growth of TSME focals?

**Author Contributions:** Conceptualization, S.I.G. and J.H.R.L.; methodology, S.I.G., A.R. and C.F.; software, S.I.G. and A.R.; validation, S.I.G., A.R., C.F., N.J.B.K. and J.H.R.L.; formal analysis, S.I.G. and C.F.; investigation, S.I.G.; resources, S.I.G.; data curation, S.I.G.; writing—original draft preparation, S.I.G.; writing—review and editing, S.I.G., A.R., C.F., N.J.B.K. and J.H.R.L.; visualization, S.I.G.; supervision, J.H.R.L.; project administration, S.I.G.; funding acquisition, S.I.G. and A.R. All authors have read and agreed to the published version of the manuscript.

**Funding:** During this project, S.I.G. was funded by the Department of Biology at the University of Washington, and A.R. was funded through grants from the Alfred P. Sloan Foundation and the Gordon & Betty Moore Foundation to the University of Washington eScience Institute. The Pacific Northwest Permanent Sample Plot Program is funded by the National Science Foundation, grant numbers LTER8 DEB-2025755 (2020–2026) and LTER7 DEB-1440409 (2012–2020).

**Institutional Review Board Statement:** Not applicable.

**Informed Consent Statement:** Not applicable.

**Data Availability Statement:** The data presented in this study are openly available on Zenodo at doi:10.5281/zenodo.5512791, reference number [21].

**Acknowledgments:** We thank the UW eScience Incubator program and the maintainers of the UW Hyak supercomputer system. We also thank Joe Ammirati, Tony Cannistra, Aji John, Rubén Manzanedo, Kavya Pradhan, and Meera Lee Sethi for feedback on project development. Tree growth data were provided courtesy of the Pacific Northwest Permanent Sample Plot Program, in partnership with the HJ Andrews Experimental Forest and Long Term Ecological Research (LTER) program, which are administered cooperatively by the USDA Forest Service Pacific Northwest Research Station, Oregon State University, and the Willamette National Forest.

**Conflicts of Interest:** The authors declare no conflict of interest. The funders had no role in the design of the study; in the collection, analyses, or interpretation of data; in the writing of the manuscript, or in the decision to publish the results.

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
