# Peer review of "Regularized Regression: A New Tool for Investigating and Predicting Tree Growth"

_forests, doi:10.3390/f12091283_

Round 1
Reviewer 1 Report
Review of forests-1381492
Regularized regression: a new tool for investigating and predicting tree growth
I read this article with great pleasure. It explains the principles of lasso regression and neighbourhood models for tree growth in a very clear way, such that I could get a good understanding of it, without being a specialist in the area. The paper provides a very interesting case study in the performance of regularized regression as compared to more complex and laborious modelling methods like conventional model selection when both methods are applied to neighbourhood modelling. The work appears very relevant and useful. I have only a few minor queries.
L206 Why log exponential? The model seems simply exponential.
Section 2.5 I did not quite understand whether you complete separated your test data set (out-of-sample) from the training data set (that was split in folds for cross validation) or whether you tested with a subset of the training data. I suppose you used a complete separate "naieve" test dataset, but I do not quite get that insight from section 2.5. I may be glossing over something or perhaps it could be stated more explicitly. Please check.
L499-501 Sentence is not finished.
Reviewer 2 Report
This excellent manuscript compares two modelling approaches with the focus on the influence of the neighborhood on the growth of 6 species in mixed stands. The authors find quite similar results with both approaches, whereas the classical methodology needs tremendously higher computational time than the regularized regression. The paper is well written, and conclusive. Thus, I have only three suggestions for possible improvements:
1st, I miss a table, giving means and measures for the variation of the main variables: Species proportions (by basal area), density, mean numbers and quadratic mean diameters per plot at least of the focal species.
2nd, In lines 160 and following, I understand that the selection of the focal trees for the four sets was done across the plots. How has this selection been done? Randomly, systematically, stratified by elevation? Please give the means as above by the data sets.
3rd, The Tables 15 and 16 should be omitted. One sentence describing the fact that the growth of focal trees is well influenced by the neighboring trees and their species identity in all data sets and in both methods is sufficient.
